# Role of Neuroglobin in the Neuroprotective Actions of Estradiol and Estrogenic Compounds

**DOI:** 10.3390/cells10081907

**Published:** 2021-07-27

**Authors:** George E. Barreto, Andrew J. McGovern, Luis M. Garcia-Segura

**Affiliations:** 1Department of Biological Sciences, University of Limerick, V94 T9PX Limerick, Ireland; George.Barreto@ul.ie (G.E.B.); Andrew.McGovern@ul.ie (A.J.M.); 2Instituto Cajal, Consejo Superior de Investigaciones Científicas (C.S.I.C.), 28002 Madrid, Spain

**Keywords:** astrocytes, estrogen receptors, microglia, mitochondria, neuron, sex differences, tibolone

## Abstract

Estradiol exerts neuroprotective actions that are mediated by the regulation of a variety of signaling pathways and homeostatic molecules. Among these is neuroglobin, which is upregulated by estradiol and translocated to the mitochondria to sustain neuronal and glial cell adaptation to injury. In this paper, we will discuss the role of neuroglobin in the neuroprotective mechanisms elicited by estradiol acting on neurons, astrocytes and microglia. We will also consider the role of neuroglobin in the neuroprotective actions of clinically relevant synthetic steroids, such as tibolone. Finally, the possible contribution of the estrogenic regulation of neuroglobin to the generation of sex differences in brain pathology and the potential application of neuroglobin as therapy against neurological diseases will be examined.

## 1. Introduction

The central nervous system (CNS) is a target for the ovarian hormone estradiol, which not only regulates the activity of the neuronal circuits controlling reproductive physiology, sexual behavior and energy homeostasis, but also of those participating in the processing of tactile, nociceptive, visual and auditory information, motor coordination, emotions, verbal skills, cognition, learning and memory. Estradiol also activates protective homoeostatic responses in neurons, astrocytes, microglia, oligodendrocytes and endothelial cells to maintain the function of all of these circuits under pathological conditions.

After CNS injury, the hormone reduces edema, oxidative stress, neuroinflammation, altered ionic balance, apoptosis, loss of cerebral blood flow and other tissue alterations that cause secondary neuronal death and contribute to expanding the damage over other CNS regions. In parallel, estradiol activates reparative mechanisms through the regulation of autophagy and the promotion of synaptic plasticity, postnatal neurogenesis and myelin repair [1]. In consequence, estradiol is neuroprotective in animal models of depression, stroke, traumatic CNS injury, multiple sclerosis, amyotrophic lateral sclerosis, Parkinson’s disease, diabetic and hypertensive encephalopathies, Alzheimer’s disease and other illnesses of the nervous system [1,2,3,4,5,6].

Both under physiological conditions and after CNS injury, estradiol regulates neuronal and glial cell function by a coordinated activation of extranuclear and nuclear signaling events [1]. Nuclear-initiated estrogen signaling is mediated by classical estrogen receptors (ERs), α (ERα) and β (ERβ), which are transcription factors expressed by neurons, astrocytes and microglia [7,8] and are widely distributed in numerous brain regions, including those that are frequently involved in neurodegenerative diseases, such as the hippocampus, cerebral cortex and the striatum [7,8]. These receptors can be coexpressed in the same individual brain cells [7] and, after being activated by the hormone, form homo or heterodimers, recruit coactivators, corepressors and other components of the transcriptional machinery and regulate transcription by binding to estrogen response elements in the promoters of target genes, or by interacting with other transcription factors in the cell nucleus [9].

Extranuclear-initiated estrogen signaling is mediated by classical ERs localized at extranuclear sites, such as at lipid rafts of the plasma membrane [10] or at the mitochondria [11], or by activating other types of ERs, such as G protein-coupled estrogen receptor-1 or Gq-coupled membrane estrogen receptor at the plasma membrane or the endoplasmic reticulum [12,13]. Extranuclear estrogen signaling results in the modification of the activity of ion channels, kinases, phosphatases and the signaling pathways of membrane-associated receptors, such as insulin-like growth factor-1 (IGF-1), Wnt and Notch signaling [1,12,13,14]. All these signaling pathways modulated by estradiol converge in the regulation of transcriptional events that are in turn coordinated with the transcriptional regulation activated by nuclear estrogen signaling [15]. Under conditions of brain injury, the activation of these nuclear and extranuclear initiated signaling mechanisms mediate the neuroprotective actions of estradiol.

The neuroprotective actions of estradiol are in part exerted through the transcriptional upregulation of neurotrophic factors and other protective molecules [1,14]. One such molecule is neuroglobin (Ngb) [16], a globin protein family member that is expressed by neurons and glial cells in the mammalian central and peripheral nervous system [17,18] and exerts neuroprotection [19,20]. Here we will review the protective actions of neuroglobin over glial and neuronal cells, focusing especially on the neuroglobin-enhancing actions of hormonal compounds as a therapeutic strategy to alleviate brain damage.

## 2. Neuroglobin Neuroprotective Actions

In neural cells, neuroglobin interacts with proteins involved in signal transduction [16] and regulates the activity of different signaling pathways, including PI3K [21], AMP-activated protein kinase (AMPK) [22] and Wnt signaling [23]. In addition, neuroglobin regulates mitochondrial dynamics [19], mitochondrial respiratory function and ATP production [24,25] and regulates cell metabolism [22]. However, the physiological consequences of all these effects of neuroglobin in the mammalian brain have been only partially explored. Nevertheless, it is known that neuroglobin promotes neurogenesis, possibly by increasing PSA-NCAM, Tuj1 and involving Wnt signaling [23,25,26] and neuritogenesis [27,28].

In aquatic mammals, such as whales and seals, neuroglobin is involved in the physiological adaptation to hypoxia when submerged [29], suggesting that in terrestrial mammals, neuroglobin may participate in the adaptation to brain hypoxia under pathological conditions. In agreement with this possibility, it was shown that neuroglobin protects neural cells against hypoxia-ischemia [24,30,31], oxidative stress [32] and oxygen/glucose deprivation [33] and reduces infarct lesion after experimental stroke [34,35,36]. Moreover, the fact that vascular endothelial growth factor (VEGF) stimulates neuroglobin through VEGFR2/Flk1 [37] may also suggest the involvement of neuroglobin in neovascularization and angiogenesis, two essential repairing cellular mechanisms that are severely disturbed in cerebral ischemia.

The investigation of the possible protective actions of neuroglobin and its molecular mechanisms has been addressed using a variety of cellular and animal models [38]. These studies have revealed that the neuroprotective actions of neuroglobin are not limited to hypoxic-ischemic insults. For instance, overexpression of neuroglobin in transgenic mice using adenoviral vectors reduces lesion size [39] and improves sensorimotor behavior [40] after traumatic brain injury (TBI) and promotes locomotor recovery after spinal cord injury [41]. In addition, neuroglobin overexpression protects neuronal cells from NMDA [42] and β-amyloid toxicity [42,43,44] and decreases behavioral deficits in mice with Alzheimer’s disease [40]. On the other hand, the expression of neuroglobin declines in the brain in Huntington’s disease mouse model and in Alzheimer’s disease patients in parallel to the increase in pathological alterations [45,46].

Mitochondria have emerged as a central component in the neuroprotective actions of neuroglobin [19,20]. Indeed, neuroglobin is physically localized in neuronal mitochondria [47] and its overexpression in primary neurons exposed to hypoxia results in the preservation of mitochondrial membrane potential and respiration, the enhancement of ATP production and reduced oxidative stress [48]. In addition, neuroglobin interacts with several mitochondrial proteins involved in the regulation of the intrinsic pathway of apoptosis, such as cytochrome-c_1_, voltage-dependent anion channel 1 (VDAC) or huntingtin (HTT) [16,49,50,51] and protects neuronal cells by inhibiting this cell death pathway [52].

The neuroprotective mechanisms of neuroglobin also include actions outside the mitochondria. For instance, oxidative stress promotes the recruitment of neuroglobin into lipid raft microdomains of the plasma membrane. There, neuroglobin interacts with flotillin-1 and α-subunits of heterotrimeric G proteins, preventing the decrease in cAMP levels induced by oxidative stress and promoting neuronal viability [32]. Neuroglobin also preserves the activity of Na^+^/K^+^ ATPases in the membrane, as it has been detected in the hippocampus in a model of transient global cerebral ischemia [53]. Na^+^/K^+^ ATPases are essential to maintain intracellular ion homeostasis and membrane excitability. Indeed, the preservation of its activity by neuroglobin after transient global cerebral ischemia is associated with reduced damage to CA1 hippocampal neurons [53]. Another localization of neuroglobin are the neuronal growth cones [54], structures involved in axonal growth and axonal regeneration. Indeed, neuroglobin has been shown to promote the regeneration of CNS axons after traumatic injury or ischemia [54,55].

The preservation of the activity of Na^+^/K^+^ ATPases by neuroglobin is associated with an increased membranous level of Na^+^/K^+^ ATPases β1 subunit in astrocytes [53], suggesting a role of these glial cells in the mechanisms of neuroprotection. This is in agreement with the fact that neuroglobin expression is enhanced in astrocytes and microglia after brain injury [53,56,57,58,59]. Furthermore, it has been shown that exogenous intravitreal injection of neuroglobin reduces the activation of microglia, the expression of inflammatory cytokines and the apoptosis of retinal ganglion cells in hypoxic retina [60]. Therefore, the control of neuroinflammation by astrocytes and microglia may participate in the neuroprotective response mediated by neuroglobin.

Astrocytes not only mediate neuroglobin neuroprotection, but they are also protected themselves by neuroglobin. Thus, neuroglobin increases the viability of astrocytes exposed to glucose deprivation or oxidative stress [61,62,63]. To better understand the importance of neuroglobin for cell survival, neuroglobin-depleted astrocytes lose mitochondrial membrane potential (ΔΨm), alongside increases in superoxide (O⁻₂) and hydrogen peroxide levels (H_2_O_2_), while attenuating the levels of two antioxidant enzymes: catalase and superoxide dismutase 2 (SOD2, mitochondrial) [64,65]. This may be indicative of loss of mitochondria integrity and function, thereby promoting cell death. In agreement with this, when knocking down neuroglobin, pAKT/AKT is downregulated in an in vitro scratch and metabolic injury model, therefore worsening cell viability [65]. Given the essential trophic and metabolic support provided by astrocytes to neurons, it should be expected that the protection exerted by neuroglobin on astrocytes should also contribute to sustain neuronal viability. In this regard, it is noticeable that neuroglobin is transported in exosomes from astrocytes to neurons [66], a mechanism that may help to coordinate the protective responses in both cell types.

Despite the benefits of neuroglobin over neuronal and glial cells post-injury, its clinical use is hampered because it is unable to cross membranes, or even penetrate through the blood–brain barrier (BBB), due to its physical properties and hydrophilic exterior. To circumvent this problem, transactivator of transcription protein (TAT) has been used to deliver neuroglobin to the brain. Indeed, rabbits treated with neuroglobin coupled with TAT transduction domain (TAT-Ngb) had improved neuronal outcome in terms of a significant reduction in apoptotic mechanisms in a model of subarachnoid hemorrhage [67]. Furthermore, TAT-Ngb fusion proteins preserved mitochondrial function upon oxidative damage by inducing a significant rise in superoxide dismutase, heme-oxygenase 1 (HO-1) and nuclear factor erythroid 2-related factor 2 (Nrf2) [68]. However, some challenges have been mentioned in the attempts made to facilitate neuroglobin delivery to the brain. For example, TAT-Ngb-FITC has been shown to penetrate cortical neurons in vitro, but a caveat of its therapeutic efficacy was the absence of fluorescent signals after 48 h [69]. This is a major downside when considering times post-injury, a critical period in neurological diseases as traumatic brain injury. Furthermore, SH-SY5Y (dopaminergic neurons) pre-treated with TAT-Ngb (400 nM–1 μM) 2 h prior to oxygen glucose deprivation (OGD) for 18 or 36 h had no improvements on cell viability [70]. On the contrary, TAT-Ngb rescued differentiated PC12 cells from OGD and inhibited mitochondrial apoptosis, possibly by modulating the Jak stat family [71]. In accordance with this in vitro study, administration of TAT-Ngb to mice 2 h prior to middle cerebral artery occlusion (MCAO) reduced infarct volume at 24 h reperfusion and decreased neuronal death by 72 h after MCAO [35]. Although promising, most studies using neuroglobin delivery approaches have been performed on brain ischemia in vivo and in vitro models, demonstrating the urgent need to implement these approaches for other brain diseases. More recently, the use of sodium hyaluronate to facilitate the delivery of neuroglobin following MCAO and 24 h reperfusion has been explored [72]. Although this nanocarrier successfully delivered neuroglobin to the brain, the potential biological implication of the rise of its cytosolic levels was not fully explored.

## 3. Interaction of Estradiol and Neuroglobin Neuroprotective Actions

The neuroprotective actions of estradiol and neuroglobin share several common mechanisms. In part, this is due to the fact that the neuroglobin gene is upregulated by estradiol in neurons [73] and glial cells [57]. The regulation of neuroglobin expression by estradiol is exerted through membrane-initiated estrogen signaling, involving ERβ activation of p38/MAPK [73], and also by direct nuclear-initiated transcriptional regulation of the neuroglobin gene by ERα and ERβ, which probably act by binding to other transcription factors, such as Sp1, activator protein 1 (AP-1) or NF-κB, given the absence of canonical estrogen response elements the neuroglobin promoter [73,74,75,76].

It is possible that the mechanisms of neuroglobin regulation by estradiol may differ between different cell types and depending on the physiological conditions. For instance, in astrocytes, the regulation of neuroglobin expression is modified by inflammatory signals. Thus, under basal conditions the induction of neuroglobin expression by estradiol in astrocytes is mainly regulated through ERβ. However, a synergic action of ERα and ERβ, which are coexpressed in astrocytes, is necessary to maintain the neuroglobin expression upregulated when NF-κB is activated by lipopolysaccharide (LPS) in these cells [57]. Another factor involved in the estrogenic regulation of neuroglobin is huntingtin (HTT), because in the mouse hippocampus and in murine striatal neurons the polyQ mutation of HTT causes the disease and impairs the upregulation of neuroglobin by estradiol [51].

In addition to increasing neuroglobin expression, estradiol promotes the accumulation of neuroglobin in the mitochondria in neuronal cells [50]. HTT is also necessary for this hormonal effect [51]. Indeed, estradiol promotes the expression of HTT and the formation of an HTT-neuroglobin complex and its translocation to the mitochondria [51]. In the mitochondria, HTT [77] and neuroglobin [49] interact with proteins involved in the control of apoptosis, such as VDAC. Through VDAC, neuroglobin inhibits the opening of mitochondrial permeability transition pore (mPTP) and the subsequent cytochrome c release [33]. Interestingly, estradiol also promotes the association of neuroglobin with cytochrome c [50]. This effect of estradiol is enhanced under conditions of oxidative stress and is one of the mechanisms activated by the hormone to prevent apoptosis. Thus, in neuroblastoma cells exposed to H_2_O_2_, the hormone further increases the mitochondrial association of neuroglobin with cytochrome c, preventing its release from the mitochondria to the cytosol and, therefore, the initiation of the apoptotic cascade [50].

Neuroglobin may also be involved in the neuroprotective actions of estradiol mediated by glial cells. Indeed, part of the neuroprotective actions of estradiol are mediated by the control of reactive gliosis and the reduction of the inflammatory response of astrocytes and microglia [78,79].

Neuroinflammatory response requires complex homeostatic regulation, because from one side neuroinflammation is an acute protective response but it may enhance neuronal damage when it is decontrolled as, for instance, under chronic neurodegenerative conditions. The regulation of neuroglobin expression in astrocytes is a good example of this complex homeostatic regulation. Thus, from one side inflammatory signaling through NF-κB, and estradiol signaling through ERβ, promote neuroglobin expression in astrocytes, probably as a protective response [57]. However, in parallel, estradiol inhibits NF-κB activation and the inflammatory response of astrocytes by a mechanism involving ERα, and, in doing so, the hormone also reduces NF-κB-mediated induction of neuroglobin [57]. These opposite actions of estradiol may be interpreted as a mechanism to maintain the level of the inflammatory response under an adequate control.

## 4. Potential Role of Neuroglobin in the Generation of Sex Differences in Brain Pathology

The estrogenic actions in the nervous system contribute to sex differences in the incidence, prevalence, age of onset or symptomatology that are observed in numerous neurodegenerative and affective disorders [8], including pathological conditions in which neuroglobin has been reported to exert neuroprotective actions in animal models. It is plausible that a sex-specific regulation of neuroglobin expression in neurons and glial cells by circulating estradiol, or by estradiol locally synthesized by brain aromatase enzyme [80], may contribute to the generation of these differences. However, this possibility has not received enough attention in the experimental designs, which usually utilize only one animal sex. Nevertheless, it has been reported that female mice have higher basal levels of neuroglobin in the hippocampus [51] and the striatum [46] than males. However, it is still unknown if these differences offer a better protection of the female hippocampus and striatum against pathological insults.

Sex differences in the expression of neuroglobin after brain injury have also been scarcely explored, although a significant increase in neuroglobin expression has been observed in the striatum of transgenic male mice affected by Huntington’s disease, but not in female animals affected by the pathology [46]. In contrast, neuroglobin mRNA levels increase in the cerebral cortex of female rats, but not in the male cortex, after ischemic stroke [81]. Since there is a significant negative correlation between the expression of neuroglobin and the volume of cortical infarct [81], sex differences in neuroglobin expression after brain injury may be potentially involved in the decreased brain infarct volume observed in young female animals in the acute phase of ischemic stroke [82].

Sex differences have also been observed in the cellular localization of neuroglobin in a mouse model of a penetrating traumatic brain injury. In this model, neuroglobin expression in the injured cerebral cortex was restricted to the tissue located in proximity to the wound and was detected in neurons, reactive astrocytes and microglia [57,59]. The area of neuroglobin expression near the wound on day 7 post-injury was similar in 2-month-old males and females. However, the cellular distribution of neuroglobin was different, with males presenting a higher colocalization of neuroglobin with the microglia marker ionized calcium binding adaptor molecule 1 (Iba1) than females [59]. A similar outcome was also observed in 5-month-old male animals under a controlled cortical impact model showing an upregulation of neuroglobin in the ipsilateral hemisphere by day 7 after injury [40]. Interestingly, a positive correlation seems to exist between the severity of the damage and the degree of neuroglobin expression in males [83]. However, the pattern of neuroglobin expression over post-TBI periods might depend on the time of assessment. Thus, a thorough evaluation of spatiotemporal neuroglobin levels is utterly necessary for a precise time course of its evolution during acute and chronic TBI periods.

The consequences of these differences in neuroglobin cellular distribution for the functional response of the cerebral cortex to injury are still unknown. Although males show a higher neuronal survival in the lesion border of the cerebral cortex than females [59], this could not be ascribed to the differences in neuroglobin localization in microglia. However, this question merits further investigation, considering that neuroglobin expression in microglia is observed after different forms of cellular injury [58,84,85] and that these cells react to pathological insults with sex-specific inflammatory, migratory and phagocytic responses [8].

## 5. Role of Neuroglobin in the Neuroprotective Actions of Natural and Synthetic Compounds with Estrogenic Activity

Health benefits have been associated with the use of several natural compounds with estrogenic activity, such as the isoflavone genistein, which is known to exert neuroprotective actions in animal models of neurodegeneration [86,87,88]. It has been reported that genistein—and another isoflavone, formononetin—increases neuroglobin expression in primary neurons [89], suggesting that neuroglobin may participate in the neuroprotective actions of phytoestrogens and that these molecules may be used as pharmacological inductors of neuroglobin.

Synthetic steroids with estrogenic activity have also been shown to induce neuroglobin expression in neural cells. Thus, it has been reported that the synthetic steroid diarylpropionitrile (DPN), a selective agonist of ERβ that exerts neuroprotective actions [90,91,92], induces the expression of neuroglobin in SK-N-BE neuroblastoma cells, while propyl pyrazole triol (PTT), a selective agonist of ERα, does not [73].

Another synthetic steroid with estrogenic actions that has been shown to upregulate neuroglobin in neural cells is tibolone. Tibolone is a selective tissue estrogenic activity regulator currently used in clinical practice for the treatment of osteoporosis and climacteric symptoms. Following administration, tibolone is converted into 3α-hydroxytibolone, 3β-hydroxytibolone and δ4-tibolone [93,94,95,96]. These metabolites are highly lipophilic, able to cross plasma membranes, hence penetrating the BBB, and act as agonists of androgen (AR), ERs and progesterone (PGR) receptors. Thus, 3α and 3β metabolites preferentially bind to ERs, mainly ERα, while δ4-tibolone binds to AR and PGR [97]. In the nervous system, tibolone exerts neuroprotective actions and promotes cognition [98,99], probably by acting in part over glial cells through the attenuation of the reactive activation of astrocytes and microglia after traumatic brain injury [100], and the regulation of astrocyte phagocytosis of brain cellular debris [101].

The induction of neuroglobin signaling is one of the neuroprotective mechanisms activated by tibolone (Figure 1). Upon binding to tibolone metabolites (α, β, and δ), ER, ARs and PGR are activated, migrate to the nucleus and interact with steroid hormone response elements localized in the promotor or upstream of *Ngb* gene. Although this genomic mechanism may stimulate the expression of neuroglobin in both glial cells and neurons, this protein can also be regulated by non-genomic signaling through activation of PI3k/AKT and p38/MAPK. Once activated, neuroglobin may interact with G protein alpha subunit (Gα) and phosphatidylinositol 3,4,5-trisphosphate 3-phosphatase (PTEN) leading to the activation of survival signaling in neurons and glial cells. Estrogenic signaling not only increases neuroglobin expression, but also induces its translocation to the mitochondria [50]. Then, neuroglobin interacts in the mitochondria with cytochrome c, mitochondrial complex proteins and scavenge ROS and NOS. Collectively, these actions carried out by neuroglobin indicate its antioxidative, anti-inflammatory, and antiapoptotic properties in the brain. Finally, by regulating these pathways, tibolone reduces the inflammatory response and increases the viability of microglia cells exposed to a metabolic insult by a mechanism involving ERβ activation, NF-κB inhibition and neuroglobin upregulation [85]. A similar mechanism, involving mainly ERβ, but also ERαmediated neuroglobin upregulation, is activated by tibolone to protect astrocytes from glucose deprivation [62]. The neuroprotective actions of tibolone are of interest, given that this compound is currently used in clinical practice. However, systemic off-target effects of tibolone may limit its use as a treatment for specific brain diseases.

## 6. Conclusions and Perspectives

Since its discovery and until the present day, neuroglobin has shown enormous potential as a therapy against inflammation and oxidative stress due to its common pathological features across many neurological diseases, including stroke, Alzheimer’s disease and traumatic brain injury, among others. The fact that in some CNS diseases, such as Alzheimer’s, the intracellular levels of neuroglobin decrease exponentially with time of injury suggests the intrinsic participation of this protein in the pathological mechanism. It should be noted that the expression of neuroglobin and its downstream signaling might be different depending on the sex, and therefore further studies exploring the sex dimorphic role of neuroglobin is expected.

Perhaps one of the greatest current challenges is to seek strategies that aim to increase both the levels and the cellular expression of neuroglobin. One strategy that has been explored in the last few years is the delivery of neuroglobin to increase its brain levels, especially in those diseases characterized by a significant reduction of endogenous neuroglobin. Considering the inability of this protein to cross membranes after brain delivery, it seems that the effect is transient and limited as its intracellular levels do not remain for a long time inside cells post-injury. Even if delivery is successful and steady, further studies are needed to delve into whether a rise in neuroglobin is directly correlated with an increased neuroprotection, so future work to unravel this possible relationship is warranted.

Finally, endogenously, neuroglobin can be stimulated by genomic and nongenomic pathways, where a wide range of pharmacological compounds have the ability to promote its expression. One of these groups capable of stimulating the transcriptional activity of *Ngb* gene in neurons and glial cells are compounds with estrogenic activity, which can be those of endogenous origin such as estradiol or of synthetic origin such as tibolone. In both cases, brain cells treated with these estrogenic compounds have a better outcome against an inflammatory stimulus, possibly by activating the endogenous synthesis of neuroglobin and ERs, since the blockade of the estrogen receptors (i.e., ERβ) is able to reduce the protection mediated by this protein. A limiting factor for a long-term therapy with estrogenic compounds is related to the side effects that can occur in other peripheral organs. Therefore, the discovery and design of more specific ligands using high-throughput screening and artificial intelligence/machine learning [102] targeting estrogen receptors only in the brain is desirable with further studies going forward in this direction.

In conclusion, regardless of the approach that may favor and stimulate neuroglobin, the neuroprotective potential of this protein is yet far from being fully explored in males and females. Unravelling which other biological functions this protein has in the brain upon injury merits to be the subject of future studies.

## Figures and Tables

**Figure 1 cells-10-01907-f001:**
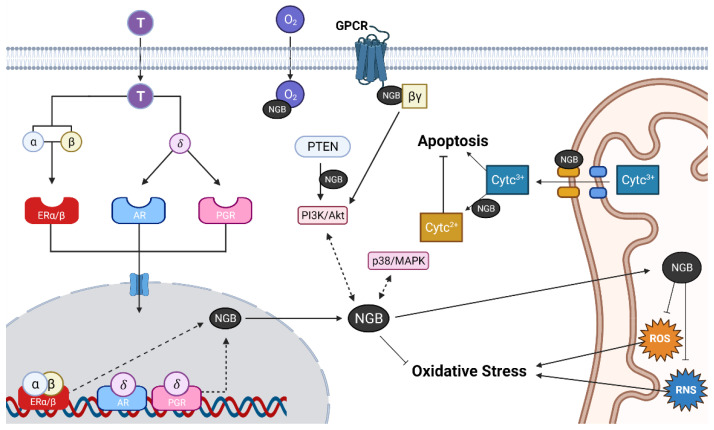
Transcriptional and post-translational regulation of neuroglobin (NGB) and biological functions. Abbreviations: T, tibolone; α, 3α-hydroxytibolone; β, 3β-hydroxytibolone; δ, δ4-tibolone; AR, androgen receptor; ERα, estrogen receptor alpha; ERβ, estrogen receptor beta; PGR, progesterone receptor; cytc, cytochrome c; βγ, G protein beta and gamma subunits; PTEN, phosphatidylinositol 3,4,5-trisphosphate 3-phosphatase; ROS, reactive oxygen species; RNS, reactive nitrogen species.

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
