# Peer review of "Role of Neuroglobin in the Neuroprotective Actions of Estradiol and Estrogenic Compounds"

_cells, 2021, doi:10.3390/cells10081907_

Round 1

Reviewer 1 Report

Titled “Role of neuroglobin in the neuroprotective actions of estradiol and estrogenic compounds”.

The manuscript is well organized and properly written. The content  is informative and the topic is interesting of current research on neuroprotection. The references list is extensive and appropriate

The authors has been made a good revision about neuroprotective actions of estradiol, neuroglobin and subsequently their common mechanisms.

I have some questions:

1.- In the section 2. “Neuroglobin neuroprotective actions” the authors point out the neuroglobin protective effects in aquatic mammals because is involved in the physiological adaptation to hypoxia when submerged. There are some available data related with the role of neuroglobin to physiological adaptation of mammals to altitude? it would be interesting to include it.

2.- In some parts of the manuscript (Lines 125-138; 172-180: 193-199; 200-209 among others) the authors refers to glia cells generically, but there are only data about astrocytes. It would be interesting to reflect some data about microglia and oligodendrocytes or change the word glia by astrocytes.

3.- line 323 misprint portential

Author Response

COMMENT

1.- In the section 2. “Neuroglobin neuroprotective actions” the authors point out the neuroglobin protective effects in aquatic mammals because is involved in the physiological adaptation to hypoxia when submerged. There are some available data related with the role of neuroglobin to physiological adaptation of mammals to altitude? it would be interesting to include it.

ACTION

The possible implication of neuroglobin in the physiological adaptation of terrestrial mammals to environmental hypoxia has been suggested, but to our knowledge it has not been demonstrated.

COMMENT

2.- In some parts of the manuscript (Lines 125-138; 172-180: 193-199; 200-209 among others) the authors refers to glia cells generically, but there are only data about astrocytes. It would be interesting to reflect some data about microglia and oligodendrocytes or change the word glia by astrocytes.

ACTION

We have revised the manuscript to replace “glia” by “astrocytes” when the sentence is dealing with this specific glial cell type. The term “glia” has been maintained when the sentence refers to glia in general. Data on microglia are presented and discussed at several places in the manuscript. To our knowledge, nothing has been published on neuroglobin and oligodendrocytes.

COMMENT

3.- line 323 misprint portential

ACTION

The spelling has been corrected.

Reviewer 2 Report

Over the past 20 years, cumulative evidence has anchored a role for estrogen and estrogenic compounds in mediating neuroprotection. However, the molecular mechanisms behind this protection are largely unknown. It is likely that many biological and environmental factors participate. This review manuscript by Barreto et al summarizes research on the role of estrogen in mediating neuroprotection through expression of Neuroglobin (NGB) and the maintenance of NGB’s intracellular levels. Although it has been known for several years that estrogen can regulate NGB expression, a summarizing review on this regulation has been missing. Therefore this review manuscript is relevant and timely. The manuscript is overall well-written and well-structured. My issues are mostly minor, however, the manuscript lacks a discussion on ER subtype expression in different cell types in the brain, a discussion on the involvement of aromatase is missing (non-the least since the authors focus to some extent on sex differences), and the clinical potential of Tibolone to increase NGB expression should be placed into context of other more selective ER agonists. Below are my specific comments, which are not listed according to importance rather to first appearance in the manuscript. Line 42-43: Reference is missing to ER subtype expression and distribution in the brain. Line 90: Remove “or”. Line 129: Should be "... increase in..." Line 133: It is not entirely correct to call the in vitro model referenced (Ref 63) as an in vitro TBI model. The model should be appropriately spelled out: In vitro Scratch and Metabolic Injury model. Line 156: Spell out MCAO when used first time Line 160: Should be "for other brain diseases", i.e. remove "the". Line 174-176: It has lately been disputed in which cell types and regions that ERa and ERb are expressed in the adult brain. This has mainly be due to issues with improper ER antibody validation. The authors should provide a short summary on the known expression of ERa and ERb in the brain (see my comment above, Line 42-43). Do all neurons express both isoforms, or do some express ERa and others ERb? Which ones? And what is the consequences for Ngb expression in such case? A reflection should be provided. Line 178: Spell out HTT Line 216: The authors write “this possibility has not received enough attention in the experimental designs”. Can the authors speculate why? Line 223-224: The authors claim that HD causes ovarian atrophy by referring to a reference of a HD mouse model. Does similar ovarian atrophy exist in human AD patients? And can this be directly linked to reduced E2 levels? Line 233-238: The cellular and spatial sex-differences in NGB distribution in the brain is interesting, and implies involvement of aromatase. The authors should include and discuss the possible involvement of aromatase. Line 243: Please consider rewording the sentence to "...a thorough evaluation of spatio-temporal neuroglobin levels ..." Line 263: Does selective ERa activation (e.g. by PPT administration) also increase NGB levels in vitro? Please clarify. Line 287-291: Tibolone activates ERa, ERb, PGR and AR signaling. What evidence exist that Tibolone modulates microglia activation solely through ERb? Also, the authors should provide a rationale for the use of Tibolone instead of selective ER agonists. The systemic off-target effects of Tiblone is likely to limit its use in specific brain diseases or injuries. Please discuss. Line 291: Which ER is more involved in NGB regulation? And in which cell type? And does ER activation induce NGB migration to the mitochondria or just the NGB expression? Please clarify or discuss. Line 295: Change "Alzheimer" to Alzheimer's disease" Line 320: Please explain how AI/machine learning may specifically help in discovering new highly selective ER ligands. Line 323: Should be "potential" Line 324: Please remove "and mediate" Line 328: Delta should be written in lower case. Line 334: Are there any supplemental material? If yes, it is missing.

Author Response

COMMENT

Line 42-43: Reference is missing to ER subtype expression and distribution in the brain.

ACTION

References have been included (lines 42-46).

COMMENT

Line 90: Remove “or”.

Line 129: Should be "... increase in..."

Line 133: It is not entirely correct to call the in vitro model referenced (Ref 63) as an in vitro TBI model. The model should be appropriately spelled out: In vitro Scratch and Metabolic Injury model.

Line 156: Spell out MCAO when used first time

Line 160: Should be "for other brain diseases", i.e. remove "the".

ACTION

All these points have been addressed following the indications of the reviewer.

COMMENT

Line 174-176: It has lately been disputed in which cell types and regions that ERa and ERb are expressed in the adult brain. This has mainly be due to issues with improper ER antibody validation. The authors should provide a short summary on the known expression of ERa and ERb in the brain (see my comment above,

Line 42-43). Do all neurons express both isoforms, or do some express ERa and others ERb? Which ones? And what is the consequences for Ngb expression in such case? A reflection should be provided.

ACTION

We have elaborated further the text on cell types and anatomical regions expressing neuroglobin in the brain as well as on the existence and implication of the coexpression of the two ER isoforms in the same brain cells (paragraph starting at line 40).

COMMENT

Line 178: Spell out HTT

ACTION

HTT has been spelled out.

COMMENT

Line 216: The authors write “this possibility has not received enough attention in the experimental designs”. Can the authors speculate why?

ACTION

The main cause is the in most studies only one animal sex is included in the experimental design. This has been mentioned in the revised manuscript (lines 224-225).

COMMENT

Line 223-224: The authors claim that HD causes ovarian atrophy by referring to a reference of a HD mouse model. Does similar ovarian atrophy exist in human AD patients? And can this be directly linked to reduced E2 levels?

ACTION

The sentence on ovarian atrophy in the HD model has been deleted given the irrelevance of this finding for neurodegenerative diseases in humans.

COMMENT

Line 233-238: The cellular and spatial sex-differences in NGB distribution in the brain is interesting, and implies involvement of aromatase. The authors should include and discuss the possible involvement of aromatase.

ACTION

In the revised manuscript we have mentioned the possible implication of brain aromatase in the generation of sex differences in neuroglobin expression (lines 221-223). We agree that this is an important issue. However, in the total absence of experimental studies assessing this question, we consider that a further discussion on the possible roles of the enzyme (which is not only expressed by neurons but also by reactive astrocytes under conditions of brain injury) will be fully speculative.

 COMMENT

Line 243: Please consider rewording the sentence to "...a thorough evaluation of spatio-temporal neuroglobin levels ..."

ACTION

The sentence has been reorganized.

COMMENT

Line 263: Does selective ERa activation (e.g. by PPT administration) also increase NGB levels in vitro? Please clarify.

ACTION

The effect of PPT has been mentioned (lines 274-275).

COMMENT

Line 287-291: Tibolone activates ERa, ERb, PGR and AR signaling. What evidence exist that Tibolone modulates microglia activation solely through ERb?

ACTION

The effect of tibolone on microglia activation is blocked by an ERb antagonist but not by an ERa antagonist (reference 86).

COMMENT

Also, the authors should provide a rationale for the use of Tibolone instead of selective ER agonists. The systemic off-target effects of Tiblone is likely to limit its use in specific brain diseases or injuries. Please discuss.

ACTION

The effects of tibolone on neuroglobin are of interest because this compound it is in use in clinical practice in a significant number of women around the world. The limitations of tibolone have been mentioned in the revised manuscript (see lines 304-306).

COMMENT

Line 291: Which ER is more involved in NGB regulation? And in which cell type?

ACTION

This has not been clarified yet and the available information is quite fragmentary and restricted to a few neuronal and glial cell lines. A systematic analysis of the role of ER subtypes in the estrogenic regulation of specific neuronal, astroglia and microglia cells from different brain regions is still needed. We have mentioned the few studies conducted with selective ER ligands through the manuscript. We have also indicated that it is possible that the mechanisms of neuroglobin regulation by estradiol may differ between different cell types and depending on the physiological conditions (paragraph starting at line 178).

QUESTION

And does ER activation induce NGB migration to the mitochondria or just the NGB expression? Please clarify or discuss.

ACTION

We have mentioned that estrogenic signaling not only increases neuroglobin expression, but also induces its translocation to the mitochondria (lines 295-296).

COMMENT

Line 295: Change "Alzheimer" to Alzheimer's disease"

ACTION

It has been changed as indicated by the reviewer.

COOMENT

Line 320: Please explain how AI/machine learning may specifically help in discovering new highly selective ER ligands.

ACTION

A reference on the possibilities offered by AI/machine learning on drug design has been quoted (line 335, reference 103).

COMMENT

Line 323: Should be "potential"

Line 324: Please remove "and mediate"

Line 328: Delta should be written in lower case.

Line 334: Are there any supplemental material? If yes, it is missing.

ACTION

All these editing points have been corrected. Mention to supplementary material has been deleted.

Reviewer 3 Report

  • A brief summary.

The proposed review article is a comprehensive summary of the knowledge about the neuroglobin function at neuronal tissue cells. The most crucial neuroprotective actions of this protein are presented. Moreover, the estradiol and neuroglobin interactions are discussed in the context of potential cellular signaling pathways or gene transcription induction, which translates into an image of sex differences in brain pathology. The work fits well with the subject of the Special Issue "Neuroglobin from Brain Protection to Cancer Progression".

  • Broad comments.

Strength:

It has to be pointed that the proposed review article possess crucial advantages:

  1. The article gets a general overview of research history, recent data, and future perspectives of the role of neuroglobin in neuroprotective action in light of potential therapy against neuroinflammation and oxidative stress.
  2. Well text organization, where discuses subjects are divided to clear chapters.
  3. Substantial references section with 100 citations.

Weakness.

For a better understanding of the estradiol (estrogenic compounds) cell influence and sex differences, the estradiol receptors concentration on cell types (neurons, astrocytes, microglia) as well at brain regions (e.g. hippocampus, cerebral cortex, striatum) should be discussed more detailed.

  • Specific comments

Line 47-49 and line167 – 170 – some of the information are duplicated:

“In addition, ERα and ERβ interact with other transcription factors in the cell nucleus, such as cAMP response element binding protein (CREB), nuclear factor kappa-light-chain-enhancer of activated B cells (NFκB), transcription factor Sp1, or signal transducer and activation of transcription (STAT3) [7].”

"The regulation of neuroglobin expression by estradiol is exerted through membrane-initiated estrogen signaling, involving ERβ activation of p38/MAPK [71], and also by direct nuclear-initiated transcriptional regulation of the neuroglobin gene by ERα and ERβ, which probably act by binding to other transcription factors, such as Sp1, AP-1 or NFκB, given the absence of canonical ERE sites in the neuroglobin promoter [71-74].”;

Line 49: Sp1 - – no abbreviation is present in the main text;

Line 127 – “…and is involved in the protective action of other factors in astrocytes [18,60,61]” – too general comment, some more information should be given;

Line174 – 177 and line 196 – 199 – repeated sense in sentences:

“Thus, under basal conditions the induction of neuroglobin expression by estradiol in astrocytes is mainly regulated through ERβ. However, a synergic action of ERα and ERβ is necessary to maintain neuroglobin expression upregulated when NFκB is activated by lipopolysaccharide (LPS) in these cells [55]”

“In primary cultures, estradiol upregulates neuroglobin in astrocytes through a mechanism involving ERβ. In addition, neuroglobin silencing prevents the anti-inflammatory effects of estradiol on astrocytes exposed to LPS [55], suggesting that neuroglobin is involved in the mechanisms elicited by estradiol to control neuroinflammation.”;

Line144 and 145: Mn-TAT PTD-Ngb and HIV TAT – no abbreviations are present in the main text;

Line 156: “…administration of 10mg/Kg” - abbreviation or typo correction is needed;

Line 156: MCAO – no abbreviation is present in the main text;

Line 170: AP-1 – no abbreviation is present in the main text;

Line 171: ERE sites – no abbreviation is present in the main text;

Line 179: “poly Q mutation” – more detailed explanation needed;

Line 221 and 231: “neuroglobin immunoreactivity” – it suggests the reaction of immune system instead of immunolabeling signal, especially if main text refer generally the inflammatory response observed e.g. in reactive glial cells;

Line 236: “microglia marker Iba1” – no abbreviation is present in the main text and there is no information on what kind of marker it is;

Line 240-241 – unclear sentence;

“For example, in male rats, severe, but not mild, TBI strongly stimulates the mRNA and protein levels of neuroglobin from 6h post-injury, reaching a peak at 20h then decreasing afterwards after 120h [82].” 

Line 276-277: suggest that Figure 1 concern only for glia cells;

Line 328: abbreviations B and Δ are not present on Figure 1;

Author Response

COMMENT

For a better understanding of the estradiol (estrogenic compounds) cell influence and sex differences, the estradiol receptors concentration on cell types (neurons, astrocytes, microglia) as well at brain regions (e.g. hippocampus, cerebral cortex, striatum) should be discussed more detailed.

ACTION

We have extended the description on these points (paragraph starting at line 40).

COMMENT

Line 47-49 and line167 – 170 – some of the information are duplicated:

“In addition, ERα and ERβ interact with other transcription factors in the cell nucleus, such as cAMP response element binding protein (CREB), nuclear factor kappa-light-chain-enhancer of activated B cells (NFκB), transcription factor Sp1, or signal transducer and activation of transcription (STAT3) [7].”

"The regulation of neuroglobin expression by estradiol is exerted through membrane-initiated estrogen signaling, involving ERβ activation of p38/MAPK [71], and also by direct nuclear-initiated transcriptional regulation of the neuroglobin gene by ERα and ERβ, which probably act by binding to other transcription factors, such as Sp1, AP-1 or NFκB, given the absence of canonical ERE sites in the neuroglobin promoter [71-74].”;

ACTION

Duplications have been eliminated.

COMMENT

Line 49: Sp1 - – no abbreviation is present in the main text;

ACTION

Abbreviation is explained.

COMMENT

Line 127 – “…and is involved in the protective action of other factors in astrocytes [18,60,61]” – too general comment, some more information should be given;

ACTION

The sentence has been deleted.

COMMENT

Line174 – 177 and line 196 – 199 – repeated sense in sentences:

“Thus, under basal conditions the induction of neuroglobin expression by estradiol in astrocytes is mainly regulated through ERβ. However, a synergic action of ERα and ERβ is necessary to maintain neuroglobin expression upregulated when NFκB is activated by lipopolysaccharide (LPS) in these cells [55]”

“In primary cultures, estradiol upregulates neuroglobin in astrocytes through a mechanism involving ERβ. In addition, neuroglobin silencing prevents the anti-inflammatory effects of estradiol on astrocytes exposed to LPS [55], suggesting that neuroglobin is involved in the mechanisms elicited by estradiol to control neuroinflammation.”;

ACTION

Duplications have been eliminated.

COMMENT

Line144 and 145: Mn-TAT PTD-Ngb and HIV TAT – no abbreviations are present in the main text;

Line 156: “…administration of 10mg/Kg” - abbreviation or typo correction is needed;

Line 156: MCAO – no abbreviation is present in the main text;

Line 170: AP-1 – no abbreviation is present in the main text;

Line 171: ERE sites – no abbreviation is present in the main text;

Line 179: “poly Q mutation” – more detailed explanation needed;

Line 221 and 231: “neuroglobin immunoreactivity” – it suggests the reaction of immune system instead of immunolabeling signal, especially if main text refer generally the inflammatory response observed e.g. in reactive glial cells;

Line 236: “microglia marker Iba1” – no abbreviation is present in the main text and there is no information on what kind of marker it is;

ACTION

Abbreviations have been explained and the editing points have been corrected.

COMMENT

Line 240-241 – unclear sentence; “For example, in male rats, severe, but not mild, TBI strongly stimulates the mRNA and protein levels of neuroglobin from 6h post-injury, reaching a peak at 20h then decreasing afterwards after 120h [82].” 

ACTION

The sentence has been deleted.

COMMENT

Line 276-277: suggest that Figure 1 concern only for glia cells;

Line 328: abbreviations B and Δ are not present on Figure 1;

ACTION

Changes have been made in the main text and the figure legend to address these points.